# A Taxonomy of Transcendence

**Natalie Abreu, Edwin Zhang,**[*] **Eran Malach**[†] **& Naomi Saphra**
Kempner Institute for the Study of Natural and Artificial Intelligence
Harvard University
{natalieabreu,ezhang}@g.harvard.edu, {emalach,nsaphra}@fas.harvard.edu

## Abstract

Although language models are trained to mimic humans, the resulting systems display capabilities beyond the scope of any one person. To understand this phenomenon, we use a controlled setting to identify properties of the training data that lead a model to transcend the performance of its data sources. We build on previous work to outline three modes of transcendence, which we call *skill denoising*, *skill selection*, and *skill generalization*. We then introduce a knowledge graph-based setting in which simulated experts generate data based on their individual expertise. We highlight several aspects of data diversity that help to enable the model's transcendent capabilities. Additionally, our data generation setting offers a controlled testbed that we hope is valuable for future research in the area.

## 1 Introduction

Language models are trained to mimic human behavioral data. This mimicry makes it tempting to anthropomorphize a system—to think of it like a person. However, not only is the model not a *person*, it is not even trained to mimic a person. Instead, the model has been trained to mimic a *group* of people with individual capacities, predilections, and biases. In some circumstances, this combination of multiple people augments their shared biases (Bender et al., 2021), but we also see the enormous advantage of training on data from a diverse set of people: often, it is possible to outperform any individual member of that group. The capacity of a generalist model to exceed individual ability is evident in a chatbot that can converse with equal competence about cryptography, international law, and the work of Dostoevsky. Our goal is to describe the circumstances in which a model, trained to mimic multiple people, is capable of **transcending** its sources by outperforming each individual.

For a group of humans, there are several ways to outperform any one individual in the group. One path is to allow the group to vote, counting each individual vote equally, thereby averaging out the bias of each individual. Another path is to route disparate knowledge by allowing a cryptographer to handle issues of cryptography and a lawyer to handle issues of international law. Finally, a group of specialists can use their shared understanding of the world as an anchor point to combine their knowledge, as when a cryptographer and a lawyer reason together through the legality of an algorithmic embargo. Like these human examples, an artificial model trained as a **generalist** can outperform the capabilities of the individual **specialists** it is trained to emulate, under the following conditions:

- **Skill denoising:** When the biases of each specialist cancel out in their average – this is classically described as "wisdom of crowds" or a voting ensemble. This scenario applies when all specialists produce data relevant to an input context, but each may make independent errors.
- **Skill selection:** When different specialists are experts on different parts of the context space. A given input may be common for one specialist and rare for another,

---

[*]Currently at OpenAI
[†]Currently at Apple

and we can reasonably assume that the specialist who frequently encounters such inputs is better equipped to handle them. By routing to the appropriate expert, the generalist model can address a wider range of topics than any individual alone. This scenario applies when the input context is familiar to at least one specialist.

- **Skill generalization:** When specialist expertise can be combined by using assumptions of shared underlying structure. Prior assumptions about how to generalize from training data allow the generalist model to interpolate, reason, and compose knowledge across domains by representing inputs in a shared semantic space. This scenario applies when the input context is unfamiliar to all individual specialists but can be understood through generalization.

In this paper, we examine the conditions under which a model can transcend its training sources, organizing them into a taxonomy of modes of transcendence. Furthermore, we connect each type of transcendence to conditions in the training data, showing that the model can transcend its sources only through a sufficiently heterogeneous set of experts. Our contributions are as follows:

- Building on definitions from Zhang et al. (2024), we formalize three modes of transcendence: skill denoising, skill selection, and skill generalization.
- We propose conditions that the training data must hold in order to achieve each mode of transcendence, and build intuition using simple theoretical settings.
- Using a synthetic knowledge graph-based setting, we empirically confirm the necessary conditions for each mode of transcendence.

We hope this will provide a useful framework for further theoretical and empirical work studying the ability of imitative models to achieve transcendent capabilities.

## 2 Definitions

### 2.1 Preliminaries

We borrow much of our notation from Zhang et al. (2024) to align with their definitions. Let $\mathcal{X} = \mathcal{T}^n$ denote the input space and $\mathcal{Y} = \mathcal{T}^m$ denote the output space over a set of tokens $\mathcal{T}$. Let $F$ be a class of functions mapping $\mathcal{X}$ to $P(\mathcal{Y})$ where $P(\cdot)$ denotes a probability distribution over the given space. Each function $f \in F$ then defines a conditional probability distribution of $y \in \mathcal{Y}$ given $x \in \mathcal{X}$ that we denote by $f(y|x)$.

Let there be $k$ experts, each associated with a conditional probability function $f_i \in F$. Let $p_1, \ldots, p_k$ denote the respective input distributions over $\mathcal{X}$ for each expert. Define the average input distribution as $\bar{p}(x) = \frac{1}{k} \sum_{i=1}^{k} p_i(x)$, and let $\text{supp}(\bar{p})$ denote its support. Define the mixture model $\bar{f}(y|x) = \sum_{i=1}^{k} g(i|x) f_i(y|x)$ where $g$ denotes the conditional probability that input $x$ is observed under expert $i$.

Then, each expert $i$ induces a distribution $\mathcal{D}_i$ over $\mathcal{X} \times \mathcal{Y}$, defined by $\mathcal{D}_i(x, y) = p_i(x) f_i(y|x)$. Let $\bar{\mathcal{D}}$ be the mixed distribution $\bar{\mathcal{D}}(x, y) = \frac{1}{k} \sum_{i=1}^{k} \mathcal{D}_i(x, y)$.

We measure the quality or skill level of each expert with a reward function $r : \mathcal{X} \times \mathcal{Y} \to \mathbb{R}$. Specifically, for a distribution $p$ over $\mathcal{X}$ and some $f \in F$, define the average reward of $f$ under $p$ as:
$$R_p(f) = \mathbb{E}_{x \sim p}\left[r_x(f)\right], \quad \text{where } r_x(f) = \mathbb{E}_{y \sim f(\cdot|x)}\left[r(x, y)\right]$$

For some fixed hypothesis class $\mathcal{H} \subseteq \{h : \mathcal{X} \to P(\mathcal{Y})\}$, the learner chooses some function $h_{\bar{\mathcal{D}}} \in \mathcal{H}$ that minimizes the expected cross-entropy with the mixture model $\bar{f}$:

$$h_{\bar{\mathcal{D}}} = \arg \min_{h \in \mathcal{H}} \mathbb{E}_{x \sim \bar{p}}[H(\bar{f}(\cdot \mid x), h(\cdot \mid x))]$$

where $H$ is the cross-entropy function.

We use the definition of "transcendence" from Zhang et al. (2024), as follows: For some test distribution $p_{\text{test}}$ over $\mathcal{X}$, we say a model achieves transcendence if

$$R_{p_{\text{test}}}(h_{\bar{\mathcal{D}}}) > \max_{i \in [k]} R_{p_{\text{test}}}(f_i)$$

We now proceed to formalize three modes of achieving transcendence.

## 2.2 Skill Denoising

Trained on noisy experts who make occasional errors, a model may outperform the experts by simply denoising their outputs. This setting is characterized by the following simplifying assumptions:

1. **_Single input distribution:_** All experts share the same distribution over $\mathcal{X}$; that is, for all $i$, we have $p_i = \bar{p}$.

2. **_In-domain test distribution:_** The support of the test distribution $p_{\text{test}}$ is contained in the support of $\bar{p}$, i.e. we have $\text{supp}(p_{\text{test}}) \subseteq \text{supp}(\bar{p})$. In other words, any example that has non-zero probability under the test distribution has non-zero probability under the training distribution.

As studied in Zhang et al. (2024), transcendence in this setting can be achieved by *low temperature sampling* as long as expert errors are uncorrelated. Low temperature sampling returns the mode of the predicted output distribution, which corresponds to a "wisdom of the crowd" approach.

## 2.3 Skill Selection

In this setting, we drop Assumption 1 from the denoising setting. Without this assumption, experts may have differing distributions over the input space. The resulting *specialist* experts each have specific *expertise*, or a subset of inputs on which they will predict a correct output. These subsets vary across experts. This setting maintains Assumption 2 that the support of the test distribution is contained in the support of the train distribution.

We claim that transcendence can be achieved when a given input context is more likely to be observed for experts that achieve a higher expected reward on that context. Specifically, the probability $g(i|x)$ of sampling expert $i$ on context $x$ is a function dependent on $x$ rather than a constant function. Intuitively, this reflects a scenario in which experts are more likely than non-experts to encounter and respond to inputs within their domain—for example, a lawyer is more likely to comment on questions of law.

As a simple example, we analyze the case of two experts.

**Theorem 2.1.** *Let a and b be two experts. For transcendence to hold, we must have that*

$$\mathbb{E}_{x \sim p_{test}} \left[ (r_x(f_a) - r_x(f_b))(g(a|x) - g(b|x)) \right] > 0$$

That is, the excess probability of seeing a given example under expert *a* compared to expert *b* is correlated with the excess reward of expert *a* on that example. We provide the proof in Appendix A.1.

## 2.4 Skill Generalization

In this setting, we drop Assumption 2, that the support of the test distribution is contained in the support of the train distribution. More strongly, we will now assume that $\text{supp}(p_{\text{test}}) \cap \text{supp}(\bar{p}) = \varnothing$; that is, the inputs at test time are never seen during training.

How can a model answer correctly if no individual expert can? If expert knowledge is representable in a shared latent space, the model can compose knowledge from different experts, producing knowledge outside the scope of any one expert's knowledge.

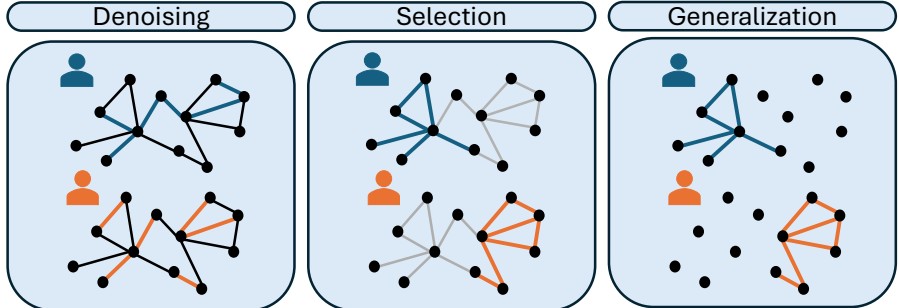

Figure 1: Illustration of the expert distributions in our knowledge-graph setting. Edges in blue and orange denote facts that the experts will respectively get correct, and the remaining edges they will get wrong. Edge opacity represents the probability of an expert generating a sample based on that edge. From left to right: (1) Denoising setting: experts have unbiased errors and uniform probability of generating each fact. (2) Selection setting: specialized experts are more likely to generate data within their expertise. (3) Generalization setting: a subcase of the setting of specialized experts. For simplicity in our generalization experiments, we set the probability of an expert generating an incorrect fact to 0.

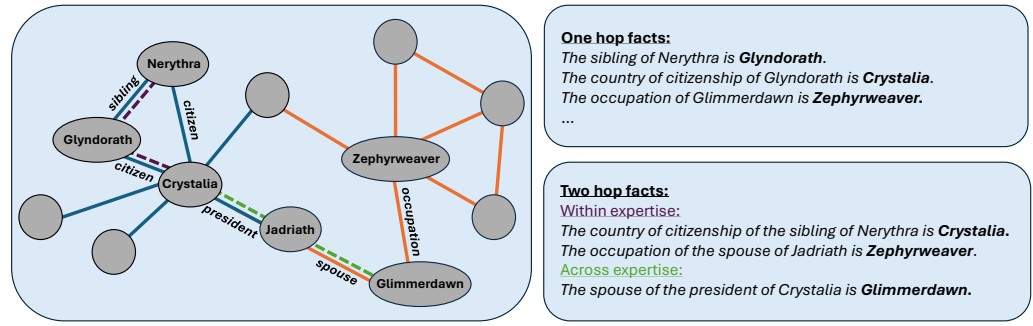

Figure 2: Example visualization of the knowledge graph with fictional entities. The blue and orange edges represent the knowledge of two different experts. A subset of nodes and edges are labeled with example entities and relations. Two-hop facts, as used in the skill generalization experiments, are split into "within expertise" (known by at least one expert) and "across expertise" (known by no expert) categories.

We evaluate this idea in a two-hop fact completion task over a knowledge graph. Each expert observes and labels a set of one-hop facts, and is able to answer certain two-hop queries composed entirely from their own knowledge. However, the test queries require combining information from multiple experts: no single expert has access to both necessary one-hop facts.

If a learner is biased toward simple solutions, and if the compositional structure of the task is simpler than memorizing all two-hop inputs, the model may generalize by composing reusable one-hop components. This enables it to answer novel two-hop queries that no individual expert could answer. We explore this mechanism more formally in Appendix A.2.

## 3 Knowledge graph setting

Our experiments use a synthetic knowledge graph and corresponding corpora of natural language sequences describing the relations in the graph. Consider the ground-truth knowledge graph $G$ consisting of a set of nodes $V$ (representing entities) and edges $E$ (representing relational facts between entities). Each fact is represented by a tuple (*head*,

*relation*, *tail*), where the *head* and *tail* are entities in the graph. We create our knowledge graph by taking the structure of the WIKIDATA-based graph from Cohen et al. (2023) and using GPT-4o-mini (OpenAI, 2024) to replace the entities with fictional names. The result is a realistically structured knowledge graph populated by facts unseen during pretraining (See Appendix B.1 for further details).

In a set of $n_e$ experts, expert $i$ has a personal knowledge graph $G_i$ that contains their knowledge of the world. Each expert is assigned a predefined amount of correct knowledge from the ground-truth graph, along with a number of incorrect beliefs.

Incorrect beliefs are modeled by introducing corrupted edges into the expert's graph. Specifically, for a fact (*head*, *relation*, *tail*) from the ground truth knowledge graph, we generate a corrupted version by replacing either the head or the tail with a different entity of the same type. Each entity in the graph is assigned a semantic type (e.g., country, person, occupation), and substitutions are restricted to entities of the same type to preserve syntactic plausibility.

For skill selection and skill generalization, we use spectral clustering over the edges to define potential areas of expertise for our synthetic experts. We split the graph into 5,000 clusters of edges. Each expert is then assigned knowledge of one or more clusters. Further details on how experts are generated in each setting are provided in the methodology sections specific to each experiment.

**Training data:** Each sample is a paragraph about a specific entity, emulating an expert writing about a particular topic. To generate a dataset of $N$ samples from a set of $n_e$ experts, we have each expert generate $N/n_e$ samples unless otherwise specified. To generate each sample, we sample a node uniformly at random from the expert's personal knowledge graph and generate a templated sentence for each edge (fact) connected to that node. Templates follow a simple pattern such as: "*The {relation} of {head} is {tail}.*" The sentences are written in random order. See Figure 2 for examples of the templated sentences.

**Evaluation:** We evaluate each model on its query completion accuracy, computed as follows. For each fact in the ground-truth set – represented as a triple *(head, relation, tail)* – we construct a query by removing the tail and prompting the model with the remaining components (e.g., "*The {relation} of {head} is __*"). A prediction is considered correct if the model's output exactly matches the ground-truth tail. For queries with multiple correct tails, we mark the output correct if it matches any correct tail. The query completion accuracy is then defined as the percentage of facts for which the model correctly outputs the tail. In other words, we evaluate the percentage of the ground truth facts the model has memorized.

**Experiment setup:** Unless otherwise specified, all experiments finetune a pretrained GPT2 model (Radford et al., 2019). Since the knowledge graph contains entities that are completely fictional, the model's pretraining phase does not include any facts from our knowledge graph. All runs use AdamW (Loshchilov & Hutter, 2019) with learning rate 0.001, weight decay 0.1, 1000 steps of warmup with a cosine decay schedule and batch size 24.

# 4 Skill denoising

When expert sources commonly make factual errors – but each makes different errors – their collective majority vote is often correct. By leveraging this "wisdom of the crowd", a model trained on their data can transcend the accuracy of any single expert by denoising their mistakes.

**Methodology** For some number of experts $n_e$, we assign all experts a shared *coverage level* $c \in [0,1]$, defined as the fraction of the knowledge graph that each expert knows correctly. To create a personal knowledge graph $G_i$ for expert $i$, we iterate over each edge in the ground-truth graph: with probability $c$, we include the correct edge, and with probability $1 - c$ we instead include a corrupted edge as described in Section 3. In our experiments, we vary the number of experts $n_e$ as a proxy for uncorrelated errors. Since errors are sampled uniformly and independently for each expert, increasing the number of experts leads to a more uniform distribution of errors in the resulting dataset.

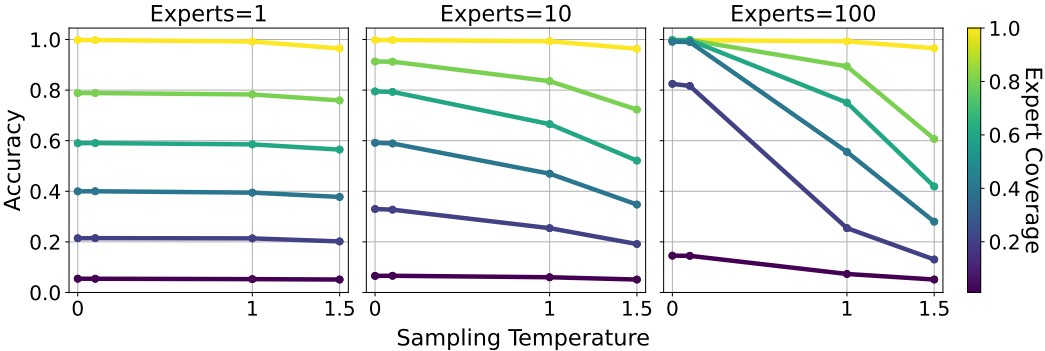

Figure 3: When trained on noisy experts, the model can leverage *low-temperature sampling* to transcend expert skill level.

**Results**   All experiments are run on 10M samples per configuration. For plots where temperature is unspecified, greedy decoding (i.e., temperature 0) is used. Accuracy refers to the query completion accuracy defined in Section 3. Consistent with our intuition, we find that with a sufficient number of experts, the model can substantially outperform individual expert skill level. Even for expert coverage scores as low as 0.2, the model achieves over 80% query accuracy when trained with 100 experts. Accuracy across different coverage levels is shown in Figure 4. As seen in Figure 3, low-temperature sampling allows the model to achieve high accuracy, while accuracy at temperature 1.0 remains close to the underlying expert coverage level.

**Takeaway**   Data diversity, in the form of uncorrelated errors, enables a model to perform at a higher skill level by leveraging the "wisdom of the crowd" with low-temperature sampling.

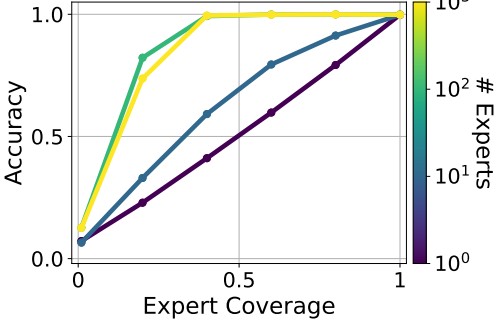

Figure 4: Expert coverage vs query accuracy on one hop facts for the denoising setting. With enough experts, even with low expert coverage, the model is highly accurate.

## 5   Skill selection

In reality, nonexperts often share their misconceptions about a topic, so their errors are rarely uncorrelated. We therefore cannot assume that a majority vote always provides transcendence through denoising. However, models can still outperform all individual experts across the training set by choosing answers from sources with the relevant expertise. For a model to transcend the experts by skill selection, it needs training data in which experts comment more on topics within their expertise than on topics where they hold common misconceptions.

**Methodology**   As described in Section 3, we partition the edges of the knowledge graph into 5,000 clusters to define potential areas of expertise. We set a coverage level $c \in [0, 1]$ shared by all experts, indicating the fraction of the graph each expert knows correctly. For each expert $i$, we generate a *coverage vector* $s_i = (s_i^{(1)}, \ldots, s_i^{(5000)})$, where each $s_i^{(j)} \in [0, 1]$ indicates expert $i$'s accuracy on edges in cluster $j$. This vector is constructed to satisfy:

$$\sum_{j=1}^{5000} s_i^{(j)} \cdot |C_j| = c \cdot |E|$$

where we denote the set of edges in cluster $j$ by $C_j$ and $|E|$ is the total number of edges in the graph. To construct expert $i$'s personal knowledge graph, we iterate over each edge

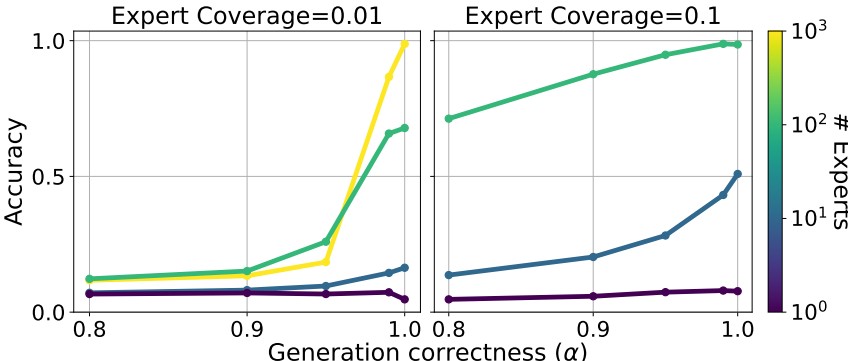

Figure 5: $\alpha$ vs query accuracy on one hop facts for the selection setting with two different coverage levels. With a sufficient number of experts and high probability that experts generate data within their "expertise", the model can achieve high accuracy even when each individual expert knows only a fraction of the knowledge graph.

$e \in E$. Let $j$ be the cluster membership of $e$, i.e., $e \in C_j$. Then with probability $s_i^{(j)}$ we include the correct edge and with probability $1 - s_i^{(j)}$ we replace it with a corrupted version.

To generate training sequences, we sample a node uniformly at random from an expert's personal knowledge graph, as in the denoising setting. However, we now introduce another parameter $\alpha \in [0, 1]$ which describes the extent to which each expert restricts commentary to their area of expertise. For each fact connected to the chosen node, we write the fact with probability $p = \alpha s_i^{(j)} + (1 - \alpha)$, where $s_i^{(j)}$ is the expert's coverage score for the cluster to which the edge belongs (i.e., the probability the edge is correct). In other words, $\alpha$ controls a weighted interpolation between the "expertise" scores and the uniform distribution. With $\alpha = 1$, a fact will be written with probability equal to the expert's level of expertise over the cluster the edge belongs to, and with lower $\alpha$, we will train on more facts outside of the expert's expertise. This provides a mechanism to control the extent to which experts write about what they know versus what they don't know.

**Results**   Each training run is on a dataset of 1M paragraphs and is trained for 10 epochs. We set two coverage levels of 0.01 and 0.1 and we vary $\alpha$ for a range of values between 0.8 and 1. Increasing the number of experts corresponds with increasing diversity of expertises. As shown in Figure 5, for both levels of coverage, the model can achieve near-perfect accuracy with a sufficient number of experts. Moreover, accuracy consistently improves as $\alpha$ increases, i.e, as experts are more likely to write about facts within their area of expertise.

**Takeaway**   When non-experts share common misconceptions about a subject (i.e. biased errors), transcendence is enabled by data diversity in the form of sources with varied expertise. To guarantee this transcendence, experts must write more frequently about topics within their domain of expertise than about those outside it.

## 6   Skill generalization

In the skill selection setting, at least one expert knows the correct answer to a question. In skill generalization, by contrast, no single expert knows the correct answer. In order to transcend through skill generalization, a model must compose knowledge from multiple experts by leveraging shared representations of the world. We study this setting using a compositional two-hop fact completion task, showing that a model can complete two-hop queries which no expert can by combining diverse expert knowledge in a latent space.

**Methodology**   We use a simplified version of the expert generation technique from Section 5 in which each expert has knowledge of a single cluster. For data generation, we

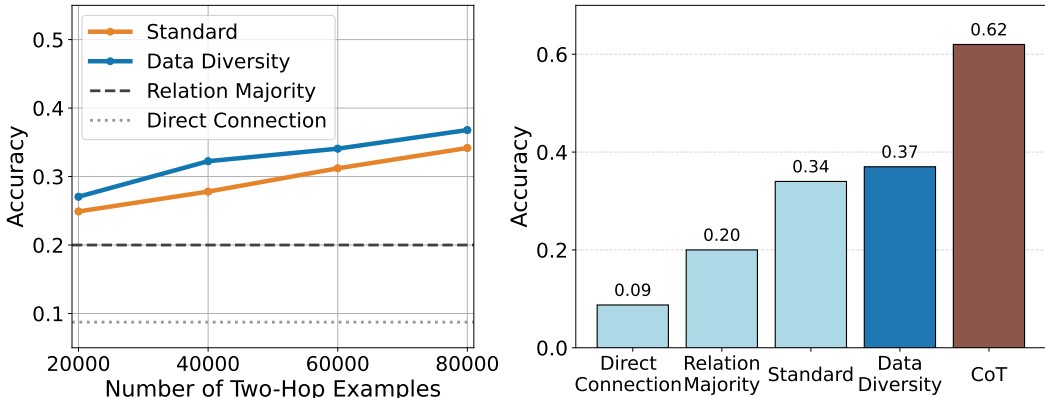

Figure 6: Left: Across-expertise two-hop query accuracy increases linearly with number of within-expertise two-hop examples seen during training. Right: Comparison of methods with full two-hop example training set in terms of across-expertise query accuracy. The standard method refers to training the model on paragraphs of one-hop facts as well as all 80,000 within-expertise two-hop sentences. For data diversity, we include diverse rephrasings of the one-hop and two-hop samples.

generate samples from each expert proportional to the size of their known cluster. As in previous experiments, the training data consists of paragraphs about entities generated from each expert's personal knowledge graph $G_i$.

To evaluate the model's ability to combine knowledge from different experts, we draw on the framework proposed by Yang et al. (2024), which measures the model's latent compositional ability via accuracy on multi-hop facts for which the model knows the relevant single hop facts. The model is said to have latent knowledge if it can correctly complete the multi-hop fact without generating the intermediate entity.

To test compositional generalization, we use *across-expertise* two-hop facts, facts where the two edges are in different clusters. By contrast, we refer to two-hop facts with both edges in the same cluster as *within-expertise* two-hop facts. (See Figure 2.) To teach the two-hop answer format, we include a set of within-expertise two-hop facts in the training data. We withhold a validation set of 6,000 within-expertise two-hop queries during training.

The across-expertise two-hop facts are unseen during training and require the model to combine specialized expertises. In addition to measuring accuracy on one-hop queries, we measure accuracy on the unseen validation set of within-expertise two-hop queries and on the test set of two-hop across-expertise queries.

We compare two-hop accuracy to baselines based on Yang et al. (2024), which describes two potential "shortcut methods" using simple co-occurrence statistics between entities and relations. We refer to these baselines as *direct connection* and *majority relation*. The direct connection baseline tests if there is a direct one-hop connection between the head and tail. The majority relation baseline measures the accuracy achieved from simply assigning the entity of the correct type (e.g., country, person, occupation) which appears most often with the second relation of the two-hop fact. (See Appendix B.3 for details).

**Experiment Details**  Since this is a more difficult task we use the 1B parameter LLaMA 3.2 model (Llama Team, 2024). However, in line with prior findings (Yang et al., 2024; Allen-Zhu & Li, 2024b), preliminary experiments in the base setting suggest that increasing model size offers limited improvement in two-hop performance. We train the model on 6M paragraphs containing one-hop facts about a given entity, combined with the training set of within-expertise two-hop facts. We repeat the two-hop facts 20 times each epoch in order to saturate accuracy on the training set and train for 10 epochs.

**Results**    Motivated by our analysis in Appendix A.2, we measure how increasing the size of the two-hop training set affects the performance on the unseen across-expertise two-hop facts. Our findings are reported in Figure 6. We find increasing the size of the training set steadily increases accuracy; by using all 80,000 within-expertise two-hop facts, the model achieves 34% accuracy on across-expertise two-hop facts, which is a nontrivial improvement over the 20% baseline. All models achieve near-perfect scores on one-hop query accuracy.

This suggests a core challenge in our setting: the number of two-hop facts known by individual experts is limited. We cannot continue to indefinitely increase the size of the training set. Therefore we explore two alternate methods: phrasing diversity and Chain-of-Thought (CoT).

**Phrasing diversity**    Inspired by Allen-Zhu & Li (2024a), we consider how augmenting data to increase the diversity of phrasing may improve model performance. Allen-Zhu & Li (2024a) argued that this type of data augmentation allowed the model to retain factual information while encouraging the model to have better latent representations rather than memorizing context. We experiment with adding phrasing diversity to both the one-hop fact paragraphs and the two-hop facts. (See Appendix B.3). We find that adding diversity to the one-hop paragraph samples increased performance from 34% to 37%, but adding diversity to the two-hop samples had no impact. We leave it to future work to explore whether more principled forms of data augmentation can enhance multihop capabilities.

**Chain-of-Thought**    Several prior works have found CoT (Wei et al., 2023) to be essential in solving knowledge manipulation tasks such as solving multihop queries (Allen-Zhu & Li, 2024b; Prystawski et al., 2023). Indeed, we see an immediate benefit of allowing the model to use CoT as it reaches over 60% accuracy on across-expertise two-hop queries (Figure 6). We note that within our framework, when the model uses CoT to explicitly name the intermediate node, it reduces the *skill generalization* problem into a *skill selection* problem. Further details on the CoT method are given in Appendix B.3.

**Takeaway**    Although skill generalization is more challenging than other forms of transcendence explored, language models can nonetheless achieve it. By increasing the diversity of surface forms or phrasing, we can promote this form of transcendence. More noticeably, we promote skill generalization by increasing the diversity of compositions provided in the training data.

## 7    Related work

**Transcendence**    Recently, Zhang et al. (2024) formally described the phenomenon of *transcendence*, in which a model outperforms the individual skills of the humans who generated its data. Zhang et al. (2024) proposed that when a model is trained on noisy data, low temperature sampling denoises the data if the noise comes from uncorrelated errors. As an example, they studied a chess model that achieves a 1500 Elo rating despite being trained to imitate players of rating 1000. Our work extends the transcendence framework to additional settings and cases where expert errors are correlated. Cunningham (2023) similarly delineated ways in which an imitative model can transcend human performance.

**Data diversity & Knowledge acquisition**    Our setup is similar to Allen-Zhu & Li (2024a), which used a synthetic biography dataset to show that data augmentation enables more flexible knowledge extraction. While they focus on extraction, we also study multi-hop composition. Zhu et al. (2025) suggested adding diverse data formats to improve knowledge acquisition. Allen-Zhu & Li (2024b) showed that chain-of-thought is critical for knowledge manipulation; we similarly observe benefits but focus on how data diversity supports implicit representations. Naik et al. (2024) studied how diversity in prompting at inference time can improve a model's reasoning ability. Chang et al. (2024) also used fictional entities to study knowledge acquisition, but focused on how knowledge is acquired over the course of training. Our finding that skill generalization relies on diverse examples of compositions

also reflects a broader conclusion in the compositionality literature concerned with data diversity (Berlot-Attwell et al., 2024; Levy et al., 2023; Oren et al., 2021; Rahimi et al., 2024).

**Knowledge composition**   Several works have studied failures of knowledge composition in Transformers (Dziri et al., 2023; Press et al., 2023; Wang et al., 2024; Yang et al., 2024; Saparov et al., 2023). Press et al. (2023) measured these failures as a *compositionality gap*, the fraction of compositional questions which are incorrectly answered despite correct answers to their atomic components. They found that this gap does not shrink with model scale. Wang et al. (2024) studied a path finding problem in which transformers must learn adjacency and reachability matrices. They found that transformers fail to learn reachability through transitive relationships, i.e. they cannot deduce a full path after seeing its segmented components. Yang et al. (2024) studied whether language models perform latent reasoning when answering questions that involve multi-hop knowledge. They intervened on each component hop to study how the model's recall changes when the prompt changes. Their findings suggest that scaling model size can promote better recall for the first hop of reasoning, but not for the second hop.

**Skill generalization**   Many works have studied the ability of neural networks to generalize beyond their training data, in particular through implicit regularization. Most relevant to our setting of skill generalization are Goldblum et al. (2024), which used information theory to argue that models are biased towards solutions with low Kolmogorov complexity, and Zadrozny (2000), which demonstrates that memorization becomes high-complexity as the corpus grows, causing compositional behavior to arise in a formal language setting.

**Ensembling & Model fusion**   In our work, a model might be seen as building an ensemble of the diverse experts who generated its training data. Because the training data reflects an implicit ensemble of experts, our results often mirror findings in the literature on model ensembling. These works have studied how to denoise, select, or combine knowledge from the model perspective – that is, how to improve performance in these settings using multiple models or agents. For instance, Wang et al. (2023) and Li et al. (2024) demonstrate how majority voting over a model's outputs can improve performance. Work on Mixture-of-Expert (MoE) models (Lepikhin et al., 2020; Fedus et al., 2022), ensembling (Liu et al., 2021; Li et al., 2022; Gururangan et al., 2023) design architectures explicitly for skill selection by combining diverse expert models, whereas we define experts that generate the training data for a simpler architecture. Work on model fusion (Wan et al., 2024; Mavromatis et al., 2024, *i.a.*) is yet another way to combine diverse models for an interpolated ensemble. Li et al. (2022) and Gururangan et al. (2023) train separate models on different clusters of documents and combine models via ensembling for inference, deliberately exploiting data diversity through model architecture.

## 8   Discussion

In this work, we outline three types of transcendence in which an imitative model can outperform the sources that generated its data. We analyze what conditions of the data must hold to allow for transcendence in each setting. In particular, we highlight several aspects of data diversity: For *skill denoising*, low temperature sampling enables transcendence as long as errors are uncorrelated. For *skill selection*, the model can accumulate knowledge from specialized experts as long as experts tend to generate data within their expertise. For *skill generalization*, we draw on the model's simplicity bias to argue that the model will learn to combine expert knowledge once the training data has high enough complexity through its diverse phrasing and examples of knowledge composition.

While valuable for isolating specific phenomena, our controlled experimental setup is limited, and we encourage future work that investigates these ideas in more real-world settings. We also encourage future work developing additional settings in which transcendence may occur – for example, our framework does not capture the idea of *skill discovery*. We hope our work is a useful starting point for broader investigation in the area.

## Acknowledgements

This work has been made possible in part by a gift from the Chan Zuckerberg Initiative Foundation to establish the Kempner Institute for the Study of Natural and Artificial Intelligence. Our work was improved by conversations with Annabelle Michael Carrell, Anat Kleiman, Benjamin L. Edelman, Milind Tambe, and Sham M. Kakade as well as with anonymous COLM reviewers. Some experiments were based on code provided by Annabelle Michael Carrell.

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

# A Proofs and Analysis

## A.1 Skill selection analysis

*Proof of Theorem 2.1.* Let $a$ and $b$ denote two experts. For transcendence to hold, we have $R_{p_{test}}(h_{\bar{D}}) > R_{p_{test}}(f_a)$ and $R_{p_{test}}(h_{\bar{D}}) > R_{p_{test}}(f_b)$. Starting with the first inequality, we have

$$
\begin{aligned}
0 < R_{p_{test}}(h_{\bar{D}}) - R_{p_{test}}(f_a) &= \mathbb{E}_{x \sim p_{test}}\left[r_x(\bar{f}) - r_x(f_a)\right] \\
&= \mathbb{E}_{x \sim p_{test}}\left[g(a|x)r_x(f_a) + (1 - g(a|x))r_x(f_b) - r_x(f_a)\right] \\
&= \mathbb{E}_{x \sim p_{test}}\left[(1 - g(a|x))(r_x(f_b) - r_x(f_a))\right] \\
&= \mathbb{E}_{x \sim p_{test}}\left[(g(b|x))(r_x(f_b) - r_x(f_a))\right]
\end{aligned}
$$

And symmetrically, we have $\mathbb{E}_{x \sim p_{test}}\left[(g(a|x))(r_x(f_a) - r_x(f_b))\right] > 0$

Combining these we get the desired result:

$$
\mathbb{E}_{x \sim p_{test}}\left[(r_x(f_a) - r_x(f_b))(g(a|x) - g(b|x))\right] > 0
$$

$\square$

## A.2 Skill generalization analysis

We develop a simple case study to provide intuition for the skill generalization setting, based on our knowledge graph-based experiment setup.

Let $G = (V, E)$ denote a knowledge graph where the nodes $V$ represent entities and the edges represent relational facts between entities. Let $R$ denote a set of relation types. Then each edge $e \in E$ is represented by a tuple $(a, r, b)$ with $a, b \in V$ and $r \in R$. Let $F^{(1)}$ denote the set of one-hop facts in $G$. That is,

$$
F^{(1)} = \{e = (a, r, b) : e \in E\}
$$

For simplicity assume there is a deterministic mapping of one-hop facts, that is, for a given prefix $(a, r)$ there is at most one label $b$. We will refer to this type of prefix $(a, r)$ as a "one-hop input."

A *two-hop fact* is represented by a tuple $(a, r_1, r_2, c)$ such that there exists a bridge entity $b \in V$ where $(a, r_1, b) \in F^{(1)}$ and $(b, r_2, c) \in F^{(1)}$. Let $F^{(2)}$ denote the set of two-hop facts:

$$
F^{(2)} = \left\{(a, r_1, r_2, c) \,\middle|\, \exists b \in V \text{ such that } (a, r_1, b), (b, r_2, c) \in F^{(1)}\right\}
$$

In particular, $G$ induces a function $f^* : V \times R \times R \to V$ that encodes the two-hop facts in the knowledge graph. We assume $f^*$ is only defined on triples $(a, r_1, r_2)$ for which such a bridge entity $b$ exists in the graph. We will refer to such a triple $(a, r_1, r_2)$ as a "two-hop input."

Then $f^*$ can be expressed as a compositional function:

$$
f^*(a, r_1, r_2) = g^*(g^*(a, r_1), r_2) \quad \text{where } g^*(a, r) = b : (a, r, b) \in F^{(1)}
$$

Let the input space $\mathcal{X}$ contain strings of the type $(a, r_1, r_2)$ such that $a \in V$, $r_1, r_2 \in R$ and $(a, r_1, r_2, c)$ is a two-hop fact in $G$ for some $c \in V$. Define the output space as $\mathcal{Y} = V \cup \{\epsilon\}$ where $\epsilon$ represents a default null label.

Now we will partition the graph according to $k$ "experts." Let $\{F_i^{(1)} : i = 1 \dots k\}$ be a partition of one-hop facts of the graph. Then define the input space of expert $i$ as

$$
\mathcal{X}_i = \left\{(a, r_1, r_2) \mid \exists b, c : (a, r_1, b), (b, r_2, c) \in F_i^{(1)}\right\}
$$

That is, $\mathcal{X}_i$ is the two-hop inputs where both relevant one-hop facts are in $F_i$. Let $p_i$ be some distribution over $\mathcal{X}_i$ such that every element of $\mathcal{X}_i$ has non-zero probability. Let each expert

have a label function $f_i : \mathcal{X}_i \to Y$. Assume that given an input $x \in \mathcal{X}_i$, expert $i$ will always provide the correct label. That is, for any $x = (a, r_1, r_2)$, $f_i(x) = c$ where $(a, r_1, r_2, c) \in F^{(2)}$. Let $D = \{(x, f_i(x)) : x \sim \mathcal{X}_i, i \sim [1, \ldots, k]\}$ such that $|D| = N$ be a training set of $N$ unique samples.

We will define a hypothesis class that contains two classes of memorizer functions: one class $\mathcal{H}_{mem}$ that uses a single lookup via a lookup table $T_h^{(2)}$ to map inputs to outputs, and a second class $\mathcal{H}_{comp}$ that uses a compositional function with a lookup table $T_h^{(1)}$. Formally, let $T_h^{(2)}, T_h^{(1)}$ be lookup tables where $T_h^{(2)}$ is size $V \times R \times R$ and $T_h^{(1)}$ is of size $V \times R$. Let $\mathcal{H} = \mathcal{H}_{mem} \cup \mathcal{H}_{comp}$, where

$$\mathcal{H}_{\text{mem}} = \left\{ h \mid h(a, r_1, r_2) = T_h^{(2)}[(a, r_1, r_2)] \text{ for some } T_h^{(2)} : V \times R \times R \to \mathcal{Y} \right\}$$

$$\mathcal{H}_{\text{comp}} = \left\{ h \mid h(a, r_1, r_2) = g\left(g(a, r_1), r_2\right), \text{ where } g(a, r) = T_h^{(1)}[(a, r)] \text{ for some } T_h^{(1)} : V \times R \to \mathcal{Y} \right\}$$

Define the complexity of a lookup table $T$ to be the number of non-null mappings:

$$\kappa(T) := |\{x \mid T(x) \neq \epsilon\}|$$

Define a *lookup function* $f$ to be a function whose only operation is a single lookup with a table $T_f$. Then define the complexity of the lookup function $\kappa(f) = \kappa(T_f)$. Assume there is a fixed overhead associated with composing functions. That is, for a composed function of the form $f(f(\cdot), \cdot)$, we define its complexity as:

$$\kappa(f(f(\cdot), \cdot)) = \kappa(f) + \kappa_{\text{comp}},$$

where $\kappa_{\text{comp}}$ is a constant representing the additional cost of composition compared to independent lookups.

Under an ERM learner, for training set $D$ we have that

$$h_D \in \arg\min_{h \in \mathcal{H}} \mathbb{E}_{(x,y) \sim D}[H(y, h(x))]$$

This setting is realizable, so we can write $h_D \in \mathcal{H}_D^*$ where $\mathcal{H}_D^* \subseteq \mathcal{H}$ is the set of hypotheses that achieve zero loss. Any $h$ that correctly memorizes the two-hop examples in $D$ will achieve zero loss. However, this gives us no guarantee as to how the function generalizes to unseen two-hop examples – namely, two-hop facts in which the first hop and the second hop belong to different expert partitions.

Here we will assume a *simplicity bias* to allow us to characterize when we will find the generalizing solution:

$$h_D \in \arg\min_{h \in \mathcal{H}_D^*} \kappa(h)$$

We now argue that, under reasonable conditions, the learner will prefer a compositional solution due to its lower complexity.

Any $h \in \mathcal{H}_{mem} \cap \mathcal{H}_D^*$ must store a separate mapping for each training example. Thus, its complexity satisfies:

$$\kappa(h) \geq |D|$$

By contrast, for any compositional function $h \in \mathcal{H}_{comp}$, the lookup table $T_h^{(1)}$ only needs to store one-hop facts. Therefore, its complexity is bounded by

$$\kappa(h) \leq |F^{(1)}| + \kappa_{comp}$$

where $\kappa_{comp}$ accounts for some fixed additional cost of composition.

This leads to a sufficient condition under which the learner prefers a compositional solution:
$|D| \geq |F^{(1)}| + \kappa_{comp}$

To understand when this condition is likely to hold, we analyze the maximum size of the training set $|D|$ for a given graph. Importantly, $|D|$ is bounded by the number of two-hop facts where both hops lie within a single expert's domain. Therefore, satisfying the condition above requires that enough such examples exist in the graph.

Let $d_{in}(v)$ and $d_{out}(v)$ denote the in-degree and out-degree of node $v$. Then the total number of one-hop facts in the graph is

$$|F^{(1)}| = \sum_{v \in V} d_{in}(v) = \sum_{v \in V} d_{out}(v)$$

Each node $v$ induces $d_{in}(v) \cdot d_{out}(v)$ two-hop facts. However, the training data consists only of two-hop facts in which both hops belong to the same expert partition $F_i^{(1)}$. The total number of such training examples is bounded by:

$$|D| \leq \sum_{i=1}^{k} |\mathcal{X}_i| = \sum_{i=1}^{k} \sum_{v \in V} d_{in}^{(i)}(v) \cdot d_{out}^{(i)}(v)$$

where $d_{in}^{(i)}(v)$ and $d_{out}^{(i)}(v)$ are the in/out degrees of node $v$ restricted to edges in partition $F_i^{(1)}$.

Combining the complexity bound and the dataset size estimate, we obtain a sufficient condition under which the learner prefers a compositional solution:

$$\sum_{v \in V} d_{in}(v) + \kappa_{comp} < \sum_{i=1}^{k} \sum_{v \in V} d_{in}^{(i)}(v) \cdot d_{out}^{(i)}(v)$$

This condition intuitively requires that enough valid two-hop facts lie within the domain of a single expert. Because expert knowledge is structured in a shared latent space, via reusable one-hop representations, these facts can be composed efficiently. When this holds, the training set can be large enough for the compositional function to be simpler than memorization – causing the learner, under a simplicity bias, to prefer generalization.

# B   Training Details

## B.1   Knowledge graph generation

For our experiments, we use a knowledge graph filled with fictional entities to generate facts the model has not seen during pretraining. We create the knowledge graph by taking the structure of the WIKIDATA-based graph from Cohen et al. (2023) and we use GPT-4o-mini (OpenAI, 2024) to replace the entities with fictional names. To generate entities, we start by asking GPT-4o-mini for a set of fictional country names to seed the fictional graph. These fictional country names are randomly assigned to replace the country entities in the original graph. We then run a BFS-like procedure in which the updated neighbors of each node are used as a context to generate the fictional name of the current node. We also provide a starting letter drawn at random to increase diversity. This guarantees that GPT-4o-mini provides a fictional entity name based only on its fictional context. The resulting knowledge graph has approximately 25,000 entities, 39 relation types, and 54,500 edges.

## B.2   Compute Resources

All training runs used 1-4 Nvidia H100s for 1-8 hours.

## B.3   Skill generalization experiment details

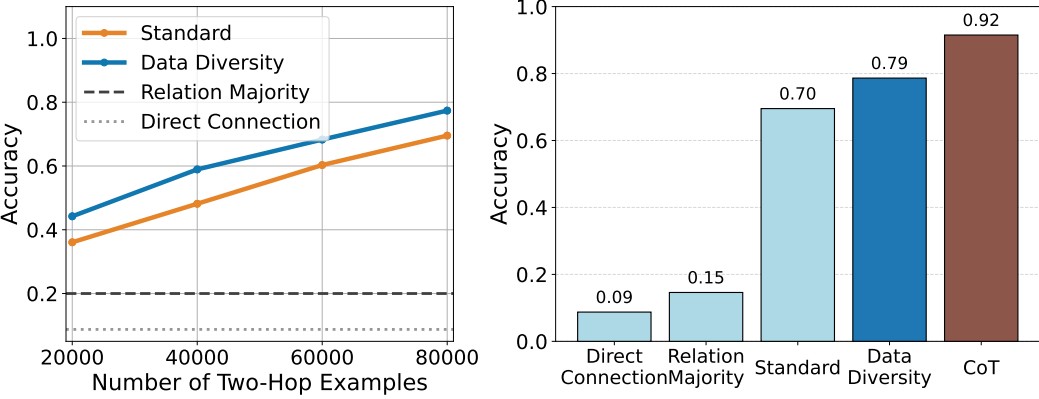

Figure 7: Left: Number of training samples vs within-expertise two-hop query accuracy on the held-out samples. Right: Comparison of methods in terms of within-expertise query accuracy.

### B.3.1   Baselines

We include baselines based on two potential "shortcuts" as described in Yang et al. (2024). For the direct connection baseline, we want to measure whether the tail entity of the two-hop fact can be predicted from frequent cooccurrences with the head entity in the training data. To do this, we measure the number of two-hop prompts in which the head-tail pair also occur in the one-hop fact training set. That is, the number of two hop prompts $(a, r_1, r_2) \mapsto c$ such that $c \in \{c : (a, r, c) \in F_{\text{one-hop}}$ for some relation $r\}$ where $F_{\text{one-hop}}$ denotes the one-hop facts in the knowledge graph. This effectively counts *all* head-tail pairs which co-occur in the one-hop facts. This accounts for 528 out of 6133 validation within-expertise two hop facts, corresponding to an accuracy of 0.086, and 5666 out of 64811 across-expertise two hop facts, corresponding to an accuracy of 0.087. We also check head-tail co-occurrences in the training set of two-hop facts and find that this corresponds to an accuracy of $< 5\%$ in both the validation and across-expertise sets, so we leave this out for clarity.

For the relation majority baseline, we measure the accuracy achieved from simply assigning the most common entity of the correct type (e.g. person, location, occupation) based on the second relation of the two-hop fact.

### B.3.2 Scores on validation set

The validation set for two-hop facts consists of roughly 6000 queries of within-expertise two-hop facts that are held out during training. Scores are reported in Figure 7.

### B.3.3 Data diversity experiments

| Level | Method | Sample |
|-------|--------|--------|
| 1 | 1 template per relation | *The place of death of Glimmerhold is Galadron Abyss. The name of the head of government of Galadron Abyss is Morrathis Voidstrider.* |
| 2 | 4 templates per relation | *Glimmerhold died in Galadron Abyss. Galadron Abyss's head of government is Morrathis Voidstrider.* |
| 3 | GPT generated | *Glimmerhold died in Galadron Abyss, which is governed by Morrathis Voidstrider.* |
| 4 | GPT generated | *Glimmerhold met its demise in the treacherous depths of Galadron Abyss, a notable locale governed by Morrathis Voidstrider. This abyss, shrouded in mystery, serves as a poignant backdrop to the tale of Glimmerhold's fate.* |

Table 1: Comparison of diversity levels in one-hop paragraph generation.

We run preliminary experiments on the usefulness of phrasing diversity in improving the model's to answer unseen two-hop facts. For each sample, we generate a version at four diversity levels:

- Level 1: Facts are written from a single template as in the standard setting. Multihop facts are written in a single template.

- Level 2: Each relation type has four templates. Each time an edge is written, one of four templates is sampled u.a.r. Multihop facts are written in a single template.

- Level 3: We use GPT-4o-mini to generate a low creativity entry based on the Level 1 paragraph. The following prompt is used: *"You will be provided with a list of facts about an entity. Your job is to write a 10-50 word encyclopedia entry about the given entity. You should not make up additional information, just rewrite the facts."*

- Level 4: We use GPT-4o-mini to generate a high creativity entry based on the Level 2 paragraph. The following prompt is used: *"You will be provided with a list of facts about an entity. Your job is to write a 10-50 word encyclopedia entry about the given entity. You should not make up additional information, just rewrite the facts. Use creative word choices and phrasing."*

To increase data diversity of the two-hop sentences, we ask GPT-4o-mini to rephrase each templated two-hop sentence. For the models marked with "Data diversity," each model was trained with 1.5 million paragraph samples, each provided at the four levels of diversity for a total of 6 million samples. Models were additionally provided with the rephrased two-hop samples repeated 20x per epoch along with the templated two-hop samples.

### B.3.4 Chain-of-Thought

For models that are allowed CoT to answer two-hop queries, we use a QA format and provide the intermediate entity before the final answer: For example, *"What is the award received by the screenwriter of Glyndor Aetheralis? Ithryndor Glaciaris; Xyphorian Starblossom."* During evaluation, we allow the model to generate an intermediate step proceeding the semicolon and judge accuracy based only on the final answer.

