# OpenReview forum: "A Taxonomy of Transcendence"
_colmweb.org/COLM/2025/Conference — COLM 2025_

### Official Review · Reviewer_vsp5 · 2025-05-08

**Rating:** 6
**Confidence:** 4
**Ethics Flag:** 1

**Summary:**

This paper studies the phenomenon of transcendence, where a system can show better performance than its knowledge sources, e.g., a set of experts. Building upon previous work studying such behavior on chess, the authors leverage knowledge graphs to sample experts and extend the study of transcendence to textual knowledge acquisition. Specifically, the authors introduce three types of transcendence: skill denoising, selection, and generalization. Experiments on fine-tuning pre-trained language models show interesting findings on how language models can perform better than simulated experts, leveraging the latent knowledge connection.

**Questions To Authors:**

1. There are many citation problems: line 70, no year. Line 141, what is the wisdom of the crowd? Line 301, use {} over the whole team’s name.

2. Lines 208-210: How is this guaranteed? Is there any probing or pre-training data analysis to show that the model does not include any facts?

3. Previous work on knowledge conflicts [1][2] shows that there is a ripple effect during knowledge acquisition or editing – one edge can influence the relevant multi-hop neighbors. Can the current evaluation framework provide extended findings considering the ripple effect?

[1] Evaluating the Ripple Effects of Knowledge Editing in Language Models

[2] “Merge Conflicts!” Exploring the Impacts of External Distractors to Parametric Knowledge Graphs

**Reasons To Accept:**

1. The paper studies an interesting topic: transcendence, where the findings might provide interesting intuition on how LLMs can achieve super expert performance, as observed in tasks such as Big-Bench Hard.

2. This paper is well-written and easy to follow. The takeaways are clear to the readers.

3.  The experiments are presented in a neat, concise, and controlled setting with the introduction of experts sampled from constructed knowledge graphs, which provides good evidence supporting the taxonomy.

**Reasons To Reject:**

1.  The motivations in the introduction can be better grounded with further citation from previous ML, NLP, or LLM literature, e.g., Line 26, “allow a group to vote” is relevant to the line of research on bagging.

2. Current findings are mostly based on training a pre-trained GPT2 (except Section 7 with Llama 3.2). It would be good to know how model parameters and pre-training design choices will impact the learning and robustness of the skill acquisition (denoising, selection, and generalization). Other newer models with similar sizes (<2B) of GPT2 can also be experimented with, e.g., Phi-1.5, Pythia, Qwen2, and SmolLM.

3. Although the analysis framework is interesting, current observations regarding diversity and CoT seem common and similar to prior findings from beyond Allen-Zhu’s line of research.

---

> ### Author Response · Authors · 2025-06-02
>
> We thank the reviewer for their time and thoughtful comments, and we provide responses below.
>
> >The motivations in the introduction can be better grounded with further citation from previous ML, NLP, or LLM literature, e.g., Line 26, “allow a group to vote” is relevant to the line of research on bagging.
>
> We appreciate the suggestion and will add further citations to the introduction.
>
> > Current findings are mostly based on training a pre-trained GPT2 (except Section 7 with Llama 3.2). It would be good to know how model parameters and pre-training design choices will impact the learning and robustness of the skill acquisition (denoising, selection, and generalization). Other newer models with similar sizes (<2B) of GPT2 can also be experimented with, e.g., Phi-1.5, Pythia, Qwen2, and SmolLM.
>
> We agree that comparing different pre-training models in our framework would be interesting. We will add experiments with more model architectures to the final version of the paper.
>
> >Although the analysis framework is interesting, current observations regarding diversity and CoT seem common and similar to prior findings from beyond Allen-Zhu’s line of research.
>
> Prior work from Allen-Zhu’s line of research shows that knowledge extraction benefits from phrasing diversity and that CoT is key to knowledge manipulation. We extend these findings by studying the multihop knowledge task in which we show that diversity of the one-hop facts can improve generalization on two-hop knowledge composition. We also connect the compositional abilities to a complexity argument which is distinct from Allen-Zhu’s work. Additionally, we posit that CoT turns an unsupervised generalisation task into an implicit skill-selection problem, which we hope to further formalize in future work. We believe that these observations along with our unifying framework contribute to prior findings and extend them into the new setting of transcendent composition across disjoint experts.
>
> >There are many citation problems: line 70, no year. Line 141, what is the wisdom of the crowd? Line 301, use {} over the whole team’s name.
>
> We appreciate your notes on this and will update the citations in the final version of the paper.
>
> > Lines 208-210: How is this guaranteed? Is there any probing or pre-training data analysis to show that the model does not include any facts?
>
> We agree that the claim needs explicit evidence. We will add a section in the appendix specifying the data generation process and add a probing analysis to guarantee the model has not seen facts prior to training. To provide more details on the dataset generation process – We start by asking GPT-4o for a set of fictional country names to “seed” the fictional graph. We then run a BFS-like procedure in which for each node, if the node has updated neighbors, those updated neighbors are used as context to generate the fictional name for the current node. This procedure guarantees that GPT-4o has not seen the original entity name when generating its fictional replacement. To further increase diversity, we also provide a starting letter drawn at random for each entity.
>
> >Previous work on knowledge conflicts [1][2] shows that there is a ripple effect during knowledge acquisition or editing – one edge can influence the relevant multi-hop neighbors. Can the current evaluation framework provide extended findings considering the ripple effect?
>
> We would also be interested in considering other use cases of our evaluation framework! We plan to release our code to make our framework available to other researchers.
>
> We thank you again for the helpful comments on our paper. We believe that we answered the main drawbacks raised in the review, and would appreciate it if you would raise your score if you believe that your concerns were addressed.

---

> > ### Comment · Reviewer_vsp5 · 2025-06-05
> >
> > Thank you for the detailed reply. I will keep my score as it is. Look forward to see the new experiments and write-ups if there is a final version.

---

> > > ### Author Response · Authors · 2025-06-09
> > > **Facts already present**
> > >
> > > To be clear about the guarantee in lines 208-210, the facts are generated and the entities in them are entirely fabricated by our data generation problem. Because the generation is novel, it is not possible for the model to have seen any of these facts prior to our training process. Examples of these novel facts are in Figure 2. We will use a standard method like [Ravichander et al. (2025)](https://arxiv.org/abs/2503.12072) to check that the entity names specifically, which are generated artificially, are not recovered from the model's training data.

---

### Official Review · Reviewer_76dv · 2025-05-12

**Rating:** 6
**Confidence:** 3
**Ethics Flag:** 1

**Summary:**

Formalization and experimentation with the concept of transcendence, where models trained on data written by experts outperform these experts. Three modes of transcendence are defined, namely skill denoising, selection and generalization. The experiments with synthetic data simulate these conditions in relational knowledge memorization and extrapolation, showing that low temperature sampling results in the phenomenon.

**Questions To Authors:**

Theorem 3.1: missing >0

**Reasons To Accept:**

Useful theoretical grounding for the types of transcendence, explained well and motivated by previous work.

Experiments demonstrate the phenomenon in a clear model system.

**Reasons To Reject:**

The knowledge graph construction uses GPT-4o, which means the "unseen" facts are still artificially likely according to GPT, so the experimental setup is biased.

Expert coverage is not clearly defined in section 5.

The dataset used is cited but no information about it is provided, nor how exactly it is anonymized by GPT-4o, making it difficult to evaluate it.

The task is somewhat simplistic since it amounts to pure relational knowledge learning + transitive closure. This is very specific but the paper overclaims the generality of the result.

The paper is very repetitive, stating the definition of the taxonomy again in every section without necessarily saying anything new about it.

Experiments use the same language model, GPT-2, raising doubt about their generalizability.

---

> ### Author Response · Authors · 2025-06-02
>
> We thank the reviewer for their time and thoughtful comments, and we provide responses below.
>
>
> >The knowledge graph construction uses GPT-4o, which means the "unseen" facts are still artificially likely according to GPT, so the experimental setup is biased.
>
> We appreciate the reviewer’s point. While GPT-4o is indeed used to generate entity names, we emphasize that these fictional names are produced via a carefully structured BFS-like process, conditioned only on a randomly chosen starting letter and previously generated fictional entities. This minimizes bias toward specific factual associations.
>
> >Expert coverage is not clearly defined in section 5.
>
> We will clarify the definition of expert coverage in section 5. Essentially, an expert with coverage c over a graph of M edges will have in expectation c*M correct facts in their personal knowledge graph.
>
> >The dataset used is cited but no information about it is provided, nor how exactly it is anonymized by GPT-4o, making it difficult to evaluate it.
>
> We will add a section in the appendix specifying the data generation process. The original dataset is a knowledge graph based off of Wikidata. Our anonymization process involves replacing the entity names of this original graph: We start by asking GPT-4o for a set of fictional country names to “seed” the fictional graph. We then run a BFS-like procedure in which for each node, if the node has updated neighbors, those updated neighbors are used as context to generate the fictional name for the current node. This procedure guarantees that GPT-4o has not seen the original entity name when generating its fictional replacement. We also provide a starting letter drawn at random to increase diversity.
>
> >The task is somewhat simplistic since it amounts to pure relational knowledge learning + transitive closure. This is very specific but the paper overclaims the generality of the result.
>
> We acknowledge that our task is a simplified setting; however, we believe that this offers a useful model system in which to explore our framework. We encourage future work to explore more general and real world settings and will be careful not to overclaim the generality of our experimental results.
>
> >The paper is very repetitive, stating the definition of the taxonomy again in every section without necessarily saying anything new about it.
>
> We appreciate the feedback, and will keep this in mind when editing the final version.
>
> >Experiments use the same language model, GPT-2, raising doubt about their generalizability.
>
> We agree that comparing different pre-training models in our framework would strengthen the generalizability of our results. For the skill generalization experiments, we report findings on Llama 3.2. We will add experiments with more model architectures to the final version of the paper.
>
> > Theorem 3.1: missing >0
>
> Thank you for catching this! We will correct Theorem 3.1.
>
> We thank you again for the helpful comments on our paper. We believe that we answered the main drawbacks raised in the review, and would appreciate it if you would raise your score if you believe that your concerns were addressed.

---

> > ### Comment · Reviewer_76dv · 2025-06-03
> >
> > Thank you for the clarifications. Assuming they will be incorporated into the paper, I raised my score to 6.
> >
> > Note that my methodological point about the bias introduced by the anonymization procedure still stands, unless proven otherwise. Namely, model performance might surpass expert performance simply because the fictional graph is not really random, but implicitly memorized by the model (not really memorized, but reflects the learned distribution of likely names in the model). Even though GPT-4o and GPT-2 are different models, they may agree on the likelihood of an anonymized triplet more than you would assume by chance.

---

### Official Review · Reviewer_nXy4 · 2025-05-12

**Rating:** 7
**Confidence:** 3
**Ethics Flag:** 1

**Summary:**

This work provides a taxonomy of ways in which language models may be able to perform better in an ensemble than any individual model, namely: skill denoising, skill selection, and skill generalization. It provides evidence of each of these, as well as investigating how factors such as number of experts impact the extent to which they occur.

**Reasons To Accept:**

* This experimentation is well-designed. While the paper only looks at a very specific task (two-hop reasoning in a knowledge graph), it carefully characterizes different ways in which models could achieve this, and then tests them directly, as well as also evaluating how number of experts might interact with these.
* The use of a knowledge graph ensures robustness and consistency that may not be present in natural data.
* The dataset avoids many of the risks of the confounds and contamination found in real data by using new names and generating text using templates (except in the 'data diversity' experiments).

**Reasons To Reject:**

Major:

* The main focus of this paper is on how to use an ensemble of language models to perform better than any of the language models individually; however,  the paper does not engage with much of the previous work on language model ensembles (for some recent examples, see, e.g., Bian et al., 2025; Huang et al., 2024; Jiang et al., 2023; Lu et al., 2024; Mavromatis et al., 2024; Xu et al., 2025; Yao et al., 2024; Yu et al., 2024). While the main focus of the present study is different in that it focuses on a specific well-defined capability and how to go about achieving it with multiple models, it is still important to consider that there are many different ways to ensemble models; and some of these works also make inferences about which factors enable ensembles to perform better.
* Even though they are clearly powerful (especially together), both the ensembling approach and the finetuning approach are straightforward and not novel (as far as I can tell), and so I am not sure that many of the formalizations (especially in Section 3.1) are necessary to put in the main paper - in my opinion, they reduce the readability.
* The discussion is extremely limited, and so the implications for theory and practical applications (and the scope of these) are unclear.
* Using GPT-4o to generate names and rephrasings runs a small risk of introducing confounds. For example, the specific process by which the entities were renamed is not stated, but if it used GPT-4o directly (e.g., by asking the model to rename specific entities), there is a possibility that there is some relation between these names and the real names that is also detectable by the GPT-2 models, though this does seem unlikely to have a strong effect.

Minor:

* Based on Appendix A1, Theorem 3.1 in the main body of the text is incomplete.
* Figure 5 is not referenced in the main text.





References:

* Bian, Y., Lin, Y., Liu, J., & Ruan, T. (2025). PToco: Prefix-based Token-level Collaboration Enhances Reasoning for Multi-LLMs. In O. Rambow, L. Wanner, M. Apidianaki, H. Al-Khalifa, B. D. Eugenio, & S. Schockaert (Eds.), *Proceedings of the 31st International Conference on Computational Linguistics* (pp. 8326–8335). Association for Computational Linguistics. https://aclanthology.org/2025.coling-main.556/
* Huang, Y., Feng, X., Li, B., Xiang, Y., Wang, H., Liu, T., & Qin, B. (2024). Ensemble Learning for Heterogeneous Large Language Models with Deep Parallel Collaboration. *Advances in Neural Information Processing Systems*, 37, 119838–119860.
* Jiang, D., Ren, X., & Lin, B. Y. (2023). LLM-Blender: Ensembling Large Language Models with Pairwise Ranking and Generative Fusion. In A. Rogers, J. Boyd-Graber, & N. Okazaki (Eds.), *Proceedings of the 61st Annual Meeting of the Association for Computational Linguistics (Volume 1: Long Papers)* (pp. 14165–14178). Association for Computational Linguistics. https://doi.org/10.18653/v1/2023.acl-long.792
* Lu, K., Yuan, H., Lin, R., Lin, J., Yuan, Z., Zhou, C., & Zhou, J. (2024). Routing to the Expert: Efficient Reward-guided Ensemble of Large Language Models. In K. Duh, H. Gomez, & S. Bethard (Eds.), *Proceedings of the 2024 Conference of the North American Chapter of the Association for Computational Linguistics: Human Language Technologies (Volume 1: Long Papers)* (pp. 1964–1974). Association for Computational Linguistics. https://doi.org/10.18653/v1/2024.naacl-long.109
* Mavromatis, C., Karypis, P., & Karypis, G. (2024, August 26). Pack of LLMs: Model Fusion at Test-Time via Perplexity Optimization. *First Conference on Language Modeling*. https://openreview.net/forum?id=5Nsl0nlStc
* Xu, Y., Chen, J., Wu, J., & Zhang, J. (2025). Hit the Sweet Spot! Span-Level Ensemble for Large Language Models. In O. Rambow, L. Wanner, M. Apidianaki, H. Al-Khalifa, B. D. Eugenio, & S. Schockaert (Eds.), *Proceedings of the 31st International Conference on Computational Linguistics* (pp. 8314–8325). Association for Computational Linguistics. https://aclanthology.org/2025.coling-main.555/
* Yao, Y., Wu, H., Liu, M., Luo, S., Han, X., Liu, J., Guo, Z., & Song, L. (2024, October 4). Determine-Then-Ensemble: Necessity of Top-k Union for Large Language Model Ensembling. *The Thirteenth International Conference on Learning Representations*. https://openreview.net/forum?id=FDnZFpHmU4
* Yu, Y.-C., Kuo, C. C., Ziqi, Y., Yucheng, C., & Li, Y.-S. (2024). Breaking the Ceiling of the LLM Community by Treating Token Generation as a Classification for Ensembling. In Y. Al-Onaizan, M. Bansal, & Y.-N. Chen (Eds.), *Findings of the Association for Computational Linguistics: EMNLP 2024* (pp. 1826–1839). Association for Computational Linguistics. https://doi.org/10.18653/v1/2024.findings-emnlp.99

---

> ### Author Response · Authors · 2025-06-02
>
> We thank the reviewer for their time and thoughtful comments, and we provide responses below.
>
> >The main focus of this paper is on how to use an ensemble of language models to perform better than any of the language models individually; however, the paper does not engage with much of the previous work on language model ensembles...
>
> Our work is in part distinguished from this literature by focusing on *data* properties that allow transcendence (rather than properties or modifications of the learner), but we agree that an overview of work in ensemble learning and multi-task learning would be valuable. We will expand the related work and add these citations.
>
> >Even though they are clearly powerful (especially together), both the ensembling approach and the finetuning approach are straightforward and not novel (as far as I can tell), and so I am not sure that many of the formalizations (especially in Section 3.1) are necessary to put in the main paper - in my opinion, they reduce the readability.
>
> We believe the formalization of each setting is necessary, in particular formalizing each setting within the framework provided by [1]. However, we will go through the formalizations and try to make the notation more readable and lightweight if possible.
>
> > The discussion is extremely limited, and so the implications for theory and practical applications (and the scope of these) are unclear.
>
> Thank you for the feedback—our discussion section will be expanded to clarify both theory and practice:
> Theory: We will add a concise subsection that
> - Clarifies how our taxonomy formalizes previously intuitive concepts, providing a framework to analyze the capabilities of an imitative model.
> - Discusses how our findings relate to existing classical results in which we (i)  link skill-denoising to classical ensemble learning under low-temperature decoding, (ii) formalize skill-selection as implicit expert-routing whenever the input-conditioned source density $g(i \mid x)$ correlates with reward, and (iii) connect our findings on compositional solutions to work on MDL and simplicity bias.
>
> Practice: We will supply concrete take-aways on the importance of data diversity that can allow an imitative model to transcend the sources it is trained on, both for understanding the capabilities of current language models and for future data curation efforts. For instance, curating data to maximize uncorrelated errors and tagging topical metadata so models can automatically learn routing without specialized architectures.
>
> Scope and limitations: We will explicitly state that the taxonomy applies when data can be partitioned into sources with distinct noise or support; tasks that require skill discovery or tool use lie beyond our present scope. We additionally acknowledge that our controlled experimental setup, while valuable for isolating specific phenomena, may differ from real-world settings.
>
> These additions will fit in one page of the camera-ready draft and eliminate ambiguity about implications.
>
> > Using GPT-4o to generate names and rephrasings runs a small risk of introducing confounds. For example, the specific process by which the entities were renamed is not stated, but if it used GPT-4o directly (e.g., by asking the model to rename specific entities), there is a possibility that there is some relation between these names and the real names that is also detectable by the GPT-2 models, though this does seem unlikely to have a strong effect.
>
> We will add a section in the appendix specifying the data generation process. We avoid providing GPT-4o with the original entity name for this reason. Specifically, we start by asking GPT-4o for a set of fictional country names to “seed” the fictional graph. We then run a BFS-like procedure in which each node’s updated neighbors are used as context to generate the fictional name for the current node. We also provide a starting letter drawn at random to increase diversity. This guarantees that GPT-4o provides a fictional entity name based only off of its fictional context.
>
> >Based on Appendix A1, Theorem 3.1 in the main body of the text is incomplete.
>
> Thank you for catching this! We will correct Theorem 3.1.
>
> > Figure 5 is not referenced in the main text.
>
> We will add a reference to Figure 5 in the skill selection experiment section.
>
> We thank you again for the helpful comments on our paper. We believe that we answered the main drawbacks raised in the review, and would appreciate it if you would raise your score if you believe that your concerns were addressed.

---

> > ### Comment · Reviewer_nXy4 · 2025-06-07
> >
> > Thank you to the authors for the detailed response, which has addressed some of my concerns. I have raised my score accordingly.
> >
> > However, I would like to note the following remaining concern:
> >
> > > Our work is in part distinguished from this literature by focusing on *data* properties that allow transcendence (rather than properties or modifications of the learner), but we agree that an overview of work in ensemble learning and multi-task learning would be valuable. We will expand the related work and add these citations.
> >
> > I agree that the focus on data properties is less widespread, but several previous works such as [C-BTM](https://arxiv.org/abs/2303.14177) and [Pack of LLMs](https://openreview.net/forum?id=5Nsl0nlStc) (which I mentioned in my initial review) do involve training on separate subsets of a larger dataset, so I do think that a more detailed discussion of the differences is warranted.

---

### Official Review · Reviewer_tCzd · 2025-05-14

**Rating:** 6
**Confidence:** 4
**Ethics Flag:** 1

**Summary:**

The paper starts with the observation that language models are trained to mimic humans and based on human-produced data but the resulting systems display capabilities beyond the scope of any one human.  For example, they have highly specialised knowledge on such a range and number of topics that no human would master. This is called transcendence.

The authors concentrate on what properties of the training data lead a model to transcend the performance of its data sources.

Their taxonomy identifies three modes of transcendence, called skill denoising, skill selection, and skill generalization. The knowledge support of the experts  is formalised as a knowledge graph whose properties can be formally manipulated.

The results show that for skill denoising, low temperature sampling enables transcendence if errors are uncorrelated. For skill selection, the model can transcend if experts tend to generate data within their expertise. For skill generalization, the authors tentatively conclude that the model will learn to combine expert knowledge once the training data has high enough complexity.

**Reasons To Accept:**

Very interesting, thought-provoking topic

Good formalisation that supports identification of precise results. This is of value beyond this specific paper.

Very clearly written.

**Reasons To Reject:**

The broader problem and goal is not defined well enough. The authors talk of ‘inhuman capacity’ (line 20). In this definition, they confuse general human capacity as capacity of the species, and capacities of an individual human, which is what they study. This confusion needs to be clarified, as they do not have a solution to specie-wise inhuman capacities.

I would have liked to see, even passingly, a comparison to previous results in ensemble learning and skill denoising, and multi-task learning, and skill generalization, which seem to reach similar conclusions.

The generalization results are not clear. What is the level of performance that will allow us to say there has been generalization?

---

> ### Author Response · Authors · 2025-06-02
>
> We thank the reviewer for their time and thoughtful comments, and we provide responses below.
>
> > The broader problem and goal is not defined well enough. The authors talk of ‘inhuman capacity’ (line 20). In this definition, they confuse general human capacity as capacity of the species, and capacities of an individual human, which is what they study. This confusion needs to be clarified, as they do not have a solution to specie-wise inhuman capacities.
>
> Thank you for catching this! This is an important distinction. We define transcendence as exceeding the capacity of the best individual human, but not as exceeding species-level capacity if you allow humans to work together. We will clarify the wording on line 20 and make sure this is clear in other definitions as well.
>
> > I would have liked to see, even passingly, a comparison to previous results in ensemble learning and skill denoising, and multi-task learning, and skill generalization, which seem to reach similar conclusions.
>
> Our work is in part distinguished from the literature by our focus on *data* properties that allow transcendence (rather than properties or modifications of the learner), but we agree that an overview of work in ensemble learning and multi-task learning would be valuable. We will expand the related work. Are there any experiments you’d want to see for comparison?
>
> > The generalization results are not clear. What is the level of performance that will allow us to say there has been generalization?
>
> We apologize for the lack of clarity in our generalization results, and will try to improve the clarity here: we define generalization as beating *both* (i) the best individual expert (0 %) and (ii) the strongest baseline (most common entity) 20% on two-hop queries absent from every expert’s support.
> In our results we achieve 34% with our standard method, which is 14% above the baseline, and with chain-of-thought we achieve 61%, in which case the gains are highly significant relative to the baseline.
>
> Why this proves generalization: Any non-zero score exceeds what any training source can produce; surpassing 20% shows the model learned compositional structure, not frequency heuristics.
>
> We will add the explicit definition above, making the evidence unambiguous.
>
>
> We thank you again for the helpful comments on our paper. We believe that we answered the main drawbacks raised in the review, and would appreciate it if you would raise your score if you believe that your concerns were addressed.

---

> > ### Comment · Reviewer_tCzd · 2025-06-04
> >
> > Thank you for your reply. Can you please elaborate on your statement
> > "Why this proves generalization: Any non-zero score exceeds what any training source can produce; surpassing 20% shows the model learned compositional structure, not frequency heuristics."?
> >
> > Why does surpassing the strongest baseline (two-hop query) numerically  directly shows compositional generalisation? How do you know that this result is not reached by other means that do not involve composition?

---

> > > ### Author Response · Authors · 2025-06-05
> > >
> > > Compositionality is generally measured by contrasting a restricted training set with a test set requiring new compositions (See [1]), do you have an alternative strategy in mind that could describe the performance or experiments you'd like to see for comparison?
> > >
> > > [1] Hupkes, Dieuwke, et al. "State-of-the-art generalisation research in NLP: a taxonomy and review." arXiv preprint arXiv:2210.03050 (2022).

---

> > > > ### Comment · Reviewer_tCzd · 2025-06-09
> > > >
> > > > Thank you for the comment. Demonstrating composition is very hard, I agree. I would like to see, for example, that the components of the composition are represented in separated spaces in the simpler pre-compositional space and more entangled in the compositional space.

---

> > > > > ### Author Response · Authors · 2025-06-10
> > > > >
> > > > > We appreciate the suggestion -- we are happy to run more experiments towards this point, though unfortunately we will not have time before the rebuttal deadline tonight. We are unable to find a straightforward way of measuring the entanglement of components from previous literature. Our results are in line with how compositionality is measured in similar works (eg [1] measure latent compositionality as "the rate of correct multi-hop answers when single-hop facts are known, without explicitly generating the intermediate answer"). However, to
> > > > > add support to the claims we make on compositionality we will explicitly outline the criteria for compositionality given by [1] and check against their "shortcut-free" criteria: that the model does not use subject-object shortcuts nor relation-object shortcuts based on co-occurrence frequencies. The relation-object shortcuts is captured by our current baseline, and we will add a baseline for subject-object shortcuts. We are happy to take further suggestions on methods as well.
> > > > >
> > > > > [1] Yang, Sohee, et al. "Do Large Language Models Perform Latent Multi-Hop Reasoning without Exploiting Shortcuts?." arXiv preprint arXiv:2411.16679 (2024).

---

### Decision · Program_Chairs · 2025-07-08

**Decision:**

Accept

**Comment:**

The paper starts with the observation that language models are trained to mimic humans and based on human-produced data but the resulting systems display capabilities beyond the scope of any one human. For example, they have highly specialised knowledge on such a range and number of topics that no human would master. This is called "transcendence".

The authors focus on on what properties of the training data lead a model to transcend the performance of its data sources.

Specifically, the authors introduce three types of transcendence: skill denoising, selection, and generalization. The knowledge support of the experts is formalized as a knowledge graph whose properties can be formally manipulated.

Experiments on fine-tuning pre-trained language models show interesting findings on how language models can perform better than simulated experts, leveraging the latent knowledge connection.
The results show that for skill denoising, low temperature sampling enables transcendence if errors are uncorrelated. For skill selection, the model can transcend if experts tend to generate data within their expertise. For skill generalization, the authors tentatively conclude that the model will learn to combine expert knowledge once the training data has high enough complexity.